# Numerical Analysis of the Racking Behaviour of Multi-Storey Timber-Framed Buildings Considering Load-Bearing Function of Double-Skin Façade Elements

**Miroslav Premrov and Erika Kozem Šilih ***

Faculty of Civil Engineering, Transportation Engineering and Architecture, University of Maribor, 2000 Maribor, Slovenia
* Correspondence: erika.kozem@um.si

**Abstract:** The paper presents an innovative approach in the modelling of multi-storey timber-framed buildings, where double-skin façade elements (DSF) are additionally considered as load-bearing wall elements against a horizontal load impact. The mathematical model with a fictive diagonal element developed for timber-framed wall elements with classical oriented strand boards (OSB) or fibre–plaster sheathing boards (FPB) is upgraded for DSF elements. The diameter of the fictive diagonal is determined with either experimental results or numerically obtained results using the time-consuming FEM model with elastic spring elements, which simulates the bonding line between the timber frame and both glazing panes. In the second part of the study, the numerical analysis of a specially selected three-storey timber-framed building was performed using the developed mathematical model with fictive diagonal elements. Two alternative calculations were performed with the DSF elements as non-resisting and racking-resisting wall elements. It was demonstrated on the selected case that the racking resistance (R) of a building can essentially increase up to 35% if DSF elements are considered as resisting wall elements. As a secondary goal of the study, it is also important to point out that by using DSF elements as racking-resisting elements, the distortion in the first floor essentially decreased. It is demonstrated on the selected numerical example that this torsional influence decreased notably (by almost 18%) when the load-bearing DSF elements were used for seismic excitation in the X direction. Therefore, such an approach can open new perspectives in designing multi-storey timber-framed buildings with a more attractive and dynamic floor plan and structure.

**Keywords:** sustainability; timber; structures; multi-storey; numerical analysis; DSF; racking resistance

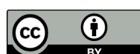

## 1. Introduction

In a sense, decreasing $CO_2$ emissions and designing energy-efficient buildings has been a topic of research since 1970. If the green house gas (GHG) emissions will continue to grow as they currently are, the Earth's average temperature will increase by at least 1.50 °C until the end of 21st century [1]. Consequently, a new strategy to design buildings with net zero emissions has to be adopted not only for new buildings, but also for a wide range of building renovations [2,3], integrating a life-cycle approach as well [4]. Timber, as a natural raw material with the potential to store $CO_2$, has the capacity to rapidly decrease GHG emissions, and seems to be the best possible solution to this problem.

A similar increasing trend may be observed in the construction of new multi-storey (MSTB) and high-rise timber buildings (HRTB). However, a "high-rise" building is mainly considered as such when surpassing 25 metres [5,6] or having more than ten storeys [7,8]. On the other hand, the term "multi-storey" tends to be a lot more defined and refers to buildings of four storeys or more, which is why it will be used in this paper. Despite the

increase and advantages shown by numerous studies in recent years, the global spread of multi-storey timber construction is still relatively low compared to massive steel construction with essentially higher lateral resistance compared to timber structures. An additional aspect is the eventual non-regularity in a multi-storey building floor-plan, which can essentially increase distortion effects in each storey and amounts to an additional limitation of multi-storey timber buildings. As a result, buildings that could be built from timber are still built from reinforced concrete or steel, which leads to an environmental performance that is poorer than if this housing stock had been built with timber—even the new timber structural systems that have been recently developed, such as those in [9].

On the other hand, the use of glazing in buildings has always contributed to openness, visual comfort, and a better daylight situation. Over the years, manufacturers have improved the thermal insulation and strength of glass [10], which enabled not only the internal illumination of buildings with large glass surfaces which were primarily south oriented, but also solar energy heating with increased solar heating gains through the transparent areas. On the other hand, the installed glazing areas that are non-load bearing in their planes in terms of assuming horizontal loads further aggravate the problem of required horizontal load-bearing capacity from a structural perspective.

Therefore, it is crucial in such cases to develop a load-bearing timber–glass wall element, which can significantly contribute to resisting the increased horizontal load impact. With their racking stiffness, such elements can increase the horizontal stiffness of the whole building and consequently decrease the torsional effects of seismic forces. In this sense, so-called single-skin timber–glass wall elements were initially developed [11], where the single-layer or thermal insulated two- or three-layered glass pane is rigidly connected to the timber frame with the bonding line [12]. It was concluded by many experimental [12–14] and numerical studies [15–17] that by using only single-skin timber–glass wall elements, the racking stiffness in particular did not increase in the expected manner and was not in the same range as the timber-framed walls with the classical sheathing boards, such as OSB or fibre–plaster boards. Therefore, special double-skin façade (DSF) timber–glass wall elements were further developed, supported by experimental [18] and further numerical studies [19]. The DSF elements were first developed primarily to be used for the energy and structural renovation of existing old buildings but can be used in new multi-storey timber buildings as well, especially in cases of a strong asymmetric position of transparent areas on the building envelope.

However, this numerical study was performed with a very time-consuming finite element model (FE) of a composite timber–glass wall element using elastic springs for simulating the bonding line between the timber frame and both glazing panes. Therefore, such an FE model is not appropriate, in practice, for any engineering application in the static or dynamical stability of multi-storey timber buildings and can be performed only in theoretical studies and analyses of a single wall element.

To avoid such time-consuming calculations in the presented study, the developed DSF timber element was integrated into a quite simple mathematical model with the fictive diagonal for simulating the racking stiffness of the bracing timber–glass wall element (Section 3). The FE model with the fictive diagonal was already developed in [20], but only for the classical fibre–plaster and OSB sheathing boards, where the sheets are stapled to the timber frame and not continuously bonded such as in the case of DSF elements. In Section 4, such modelled DSF elements are integrated into the static model of a specially selected four-storey timber-framed building with an asymmetrical position of transparent DSF elements. The influence on seismic behaviour of this selected timber building is numerically studied using an FE calculation programme SAP 2000 [21], with a special impact dedicated to the influence of the developed DSF elements to increase the racking resistance and stiffness of the building. Special attention is paid to a decrease in torsional effects caused by the seismic force if the building is analysed without and with racking resisting DSF elements.

## 2. Structural Stability of Multi-Storey Timber-Framed Buildings

### 2.1. Structural Design of Multi-Storey Timber Buildings

In the last decades, new timber products (cross-laminated timber, for instance, at the beginning of the 1990s) changed the form and especially the maximum possible height of timber buildings. In this sense, timber structures became competitive with other structures built with conventional and commonly used structural materials. Furthermore, combining timber structural elements with other commonly used building materials (brick, concrete, steel, and lately also glass) can open new perspectives on the attractive architectural forms of such hybrid timber buildings. Therefore, a combination with load-bearing glazing will be presented at the end of this section.

There are many different basic structural systems which are commonly used in multi-storey timber buildings schematically presented in Figure 1. They also differ from each other in a horizontal load transfer, and therefore allow different limits in the maximal height of timber structures. For instance, it is recommended to build timber frame-and-panel structures with up to four storeys, and a solid timber construction with CLT elements with up to ten storeys. Higher high-rise timber structures are constructed in a frame construction (14-storey Tree building in Bergen with the maximal height of 52.8 m) or even as hybrid structures combining timber primarily with a reinforced-concrete (RC) core (24-storey HoHo timber building in Vienna with the maximal height of 84 m). The limits in the height of timber buildings also strongly depend on the location of the building and the subsequent horizontal load impact (wind or earthquake).

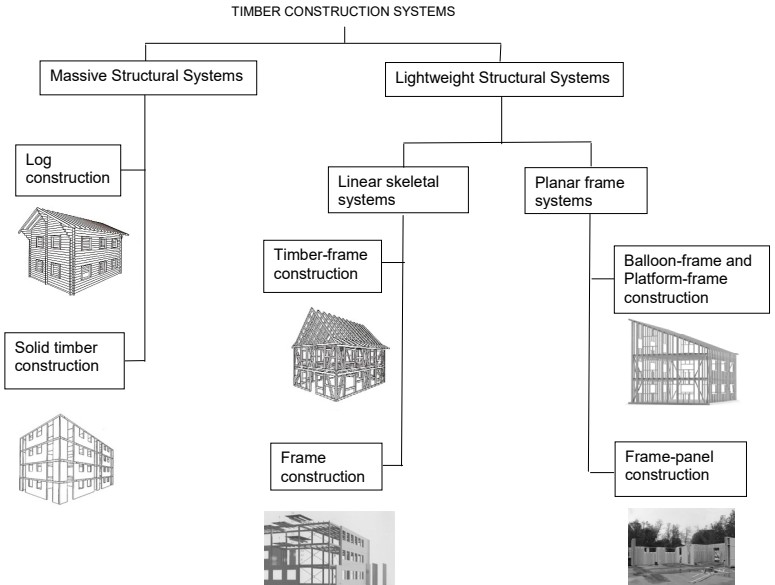

**Figure 1.** Classification of timber structural systems according to their load-bearing function.

In most cases, the more problematic of these two horizontal load actions is an earthquake, which subjects a building to a high-intensity dynamic load often resulting in catastrophic consequences. One of the basic principles when designing a building to resist seismic loads is trying to avoid plan irregularity, clearly described in, and prescribed by, the Eurocode 8 (2005) standard [22]. This means that the centre of gravity (M), where the resulting seismic force ($F_b$) acts, and the centre of rigidity/stiffness (R) of a building with a resulting horizontal resisting force should be as close to each other as possible. Unfortunately, this is an issue for energy efficient buildings with large glazing areas predominantly placed on one side of the structure, resulting in an uneven stiffness of their floor plan and an important dislocation between the centre of gravity (M) and the centre of

rigidity (R), as schematically presented in Figure 2. To avoid this distortion, it is important to consider the most external walls on the south façade as racking-resisting load-bearing elements. This means that walls with fixed glazing areas (but not windows) should also be treated as racking-resisting bracing elements and will be treated as composite elements in the timber frame-and-panel system of a timber frame and a glass sheathing, which can transmit a considerable share of horizontal forces to the basement. With such an approach, the racking resistance and stiffness of the whole analysed building can be increased, and new limits in the maximal height of multi-storey buildings constructed in a timber-framed structural system can be set. The influence of such an approach will be numerically analysed on a specially selected four-storey timber-framed building in Section 4.

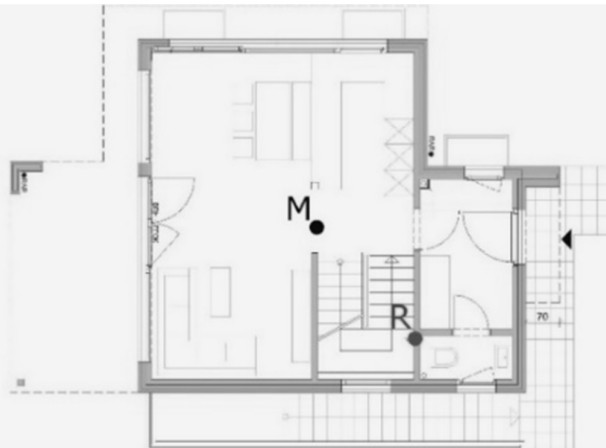

**Figure 2.** An example of dislocation between the centre of gravity (M) and the centre of rigidity (R), [16].

It must be emphasised, however, that the final structural design of a multi-storey timber-framed building additionally depends on its micro-location and height. It is generally known that according to Eurocode 1 [23], wind loads exponentially increase only from a certain height of a building upwards, while earthquake loads according to Eurocode 8 [22] increase almost linearly with the height of a building as schematically shown in Figure 3 for cases when higher loads are caused by earthquake loads. Consequently, a high horizontal load impact is particularly significant in the first storey of the building where a maximal load-bearing capacity has to be reached with all resisting wall elements. In lower stories of middle-rise or high-rise timber buildings, it is therefore of the utmost importance to consider all wall elements with fixed glass panes as resisting bracing elements, which can contribute to the overall racking resistance and stiffness of the whole building in the first storey.

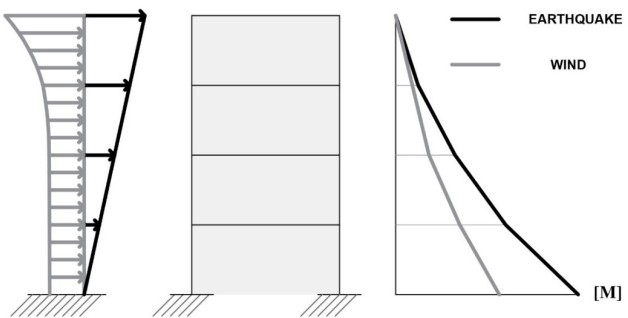

**Figure 3.** Display of how wind and earthquake load increase with the height of a building at a certain location.

*2.2. Load-Bearing Timber–Glass Wall Elements*

As mentioned before, transparent glass areas in energy-efficient timber houses are mainly installed on the south façade of the building envelope. To decrease the torsional effects of seismic forces, it is crucial to develop load-bearing timber–glass wall elements, which can contribute to the overall racking resistance and stiffness of the whole building and decrease the torsional effects. In such timber–glass wall elements, a conventional sheathing board (fibre–plaster or OSB, Figure 4b) is replaced with a glass pane (Figure 4c). The main concept of such timber–glass wall elements is that according to the basic horizontal force distribution (Figure 4a), the bonding line between the glass pane and the timber frame will take over the shear flow and the glass pane will take over the diagonal tensile force transmission (Figure 4c).

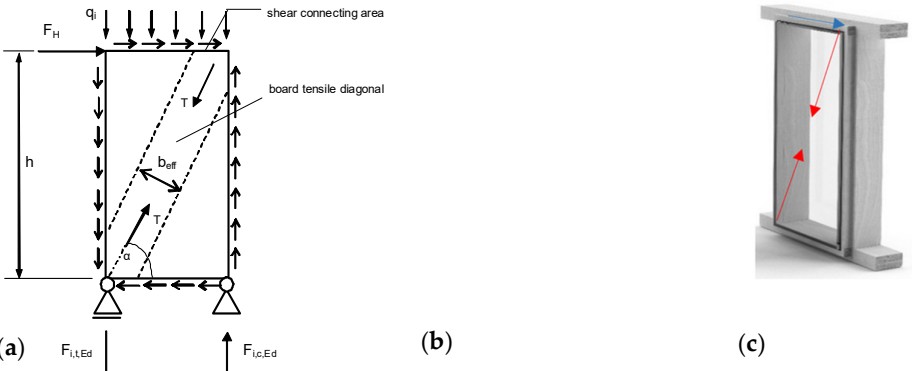

**Figure 4.** (**a**) Horizontal point load distribution in timber-framed wall element; (**b**) timber-framed wall with a conventional sheathing board; (**c**) timber–glass resisting wall element for a horizontal load impact (shear force (blue arrow), tensile diagonal force (red arrow)) [16].

In the case of timber–glass wall elements, the horizontal point load acting at the top of the element is consequently transferred to the supports in the same manner as presented in Figure 4a for any timber-framed wall elements for any sheathing boards:

- The adhesive takes over the shear stress in the gluing line;
- The tensile diagonal of the glass pane shifts the force to the support.

Timber–glass wall elements will be further separately treated as single-skin façades (SSF) with a single-glass panel and double-skin façade (DSF) elements with two panes of glazing.

2.2.1. Single-Skin Façade (SSF) Timber–Glass Wall Elements

The problem with single-skin glazing in timber-framed walls acting as racking resisting elements actually began with substantial research work in [11,13,24]. In these studies, a single-skin timber-framed load-bearing wall element with one-pane non-insulating glazing was developed using experimental and numerical tests. In [24], a special substructure was used to connect the glass pane to the timber frame. The most important technological advantage of such a type of connection is a relatively simple replacement of the glazing if it breaks. In the study of Blyberg [25], a shear wall element intended to be used as a load-bearing façade element was designed. In contradiction to [24] in this case, a non-insulating glass one-pane was rigidly bonded to a timber frame using different types of adhesives.

However, all such load-bearing timber–glass elements actually do not have any thermal insulating function and cannot be used as building envelope elements at all. Therefore, in the Wood Wisdom international research project [12], load-bearing timber–glass wall elements were further developed, where a double- or even a three-layered thermal insulating single-skin glazing was rigidly bonded directly to the timber frame structure.

Such elements can be treated as single-skin façade elements. Since various parameters (such as the type and thickness of the adhesive, the type and thickness of the glass pane, wall element dimensions, etc.) can significantly affect the racking resistance and stiffness of such timber–glass wall elements, a unique mathematical model was further developed using special spring elements to simulate the slip in the bonding line between the glass pane and the timber frame [15]. A major parametrical numerical study varying the most influential parameters stated above was further performed. It was demonstrated in many experimental and numerical studies that triple-insulating glazing can foster higher racking resistance and stiffness compared with single non-insulating glazing. However, many challenges still lie ahead to improve the type of the bonding line, the type of the used adhesive, the position of the glazing, etc., to enhance the horizontal resistance and stiffness of timber–glass wall elements, improving the structural stability of the whole timber building. The racking stiffness with a polyurethane or silicone adhesive is not in the same range as the compared timber-framed walls with conventional sheathing boards, such as OSB or fibre–plaster boards, which are prescribed by standards as primary load-bearing racking-resistant structural wall elements.

### 2.2.2. Double-Skin Façade (DSF) Timber–Glass Wall Elements

As presented in Section 2.2.1, many experimental and numerical studies highlight that the racking resistance obtained with the developed timber single-skin façade (SSF) elements is not sufficient to improve the structural behaviour of a whole building under a horizontal load impact, especially if the building is exposed to a more significant horizontal load impact as schematically presented in Figure 3. Consequently, it is crucial to develop a new load-bearing timber–glass wall element as a resisting bracing element in another way, not by using only two- or three-layer insulating single-skin (SSF) glazing, but by adding a structurally important external non-insulating single-layer glass pane. Such façade elements are treated as double-skin façade elements (DSF) and will be further presented. The introduction of innovative solutions by applying the developed DSF elements would thus expand the range of multi-storey timber building design options on account of said structural advantages. In practice, this would mean that buildings with a slightly more complex design could be built (a higher degree of floor plan or façade asymmetry, more storeys, etc., are allowed) within the scope of the same boundary conditions, and better energy efficiency could be achieved.

The schematic presentation of such a load-bearing DSF element is provided in Figure 5. It is important to point out that the thermal-insulating three-layered float glazing is placed on the internal side of the façade element and a single-layer non-insulating fully-tempered glazing on the external side. Solar shading systems can be integrated within the cavity, and the width of the cavity can vary from 200 mm to even more than 2 m. The installation of any ventilation devices, which would also be optimal for the building, is not suitable for load-bearing DSF elements. Ventilation requires openings in the load-bearing elements of the DSF system, which significantly affect the horizontal load-bearing capacity of such a DSF wall element as presented in our previous study on timber-framed wall elements with fibre–plaster sheathing boards [26]. Within the scope of the Home+ development project, the developed timber DSF wall elements constitute an additional potential of transparent areas in multi-storey timber construction [18]. In addition to the foreseen structural advantages, they provide better sound insulation of the building envelope and better energy performance as compared to the regular triple-insulating glazing [27–35].

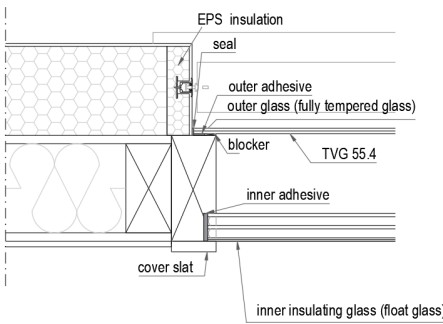

**Figure 5.** Schematic presentation of a DSF load-bearing structural wall element [19].

Recently, many studies have analysed the thermal and acoustic performance of DSF elements, but almost none of them have analysed their structural behaviour, especially in terms of determining their racking resistance. Such façade elements were in the past primarily developed with the goal of essentially improving the thermal and acoustic resistance of the building envelope [30–32]. Therefore, DSF elements have been proposed as a promising passive building technology to enhance energy efficiency and improve indoor thermal comfort [33]. Such a constructed envelope DSF element demonstrates better acoustic resistance [29] in comparison with the widely used and previously described single-skin façade (SSF) elements and can, therefore, be suitable for high-noise areas where a high level of sound insulation is required [34,35]. In wide research in Pomponi and D'Amico [28], a structural approach with a timber DSF was studied, but only for the vertical load impact. Consequently, a load-bearing DSF timber element must also be developed for a horizontal load impact to increase the possibility of also using larger transparent glass areas in multi-storey timber-framed buildings and consequently to increase the potential of wood as a natural and eco-friendly material in a variety of multi-storey residential buildings.

## 3. Mathematical Modelling of Multi-Storey Timber-Framed Buildings

### 3.1. Mathematical Modelling of Conventional Timber-Framed Wall Elements

To perform the numerical analysis, it was initially necessary to define a suitable mathematical model of the structure. For this purpose, the previously introduced mathematical model with a fictive diagonal for determining the racking stiffness of timber-framed wall elements with conventional OSB or fibre–plaster (FPB) sheathing material [20] was applied and further developed for the timber–glass wall elements stiffness simulation. It is important to point out that in the case of OSB or FPB, sheathing boards are connected to the timber frame with mechanical fasteners, usually located at a constant distance ($s_{eff}$). Consequently, the effective bending stiffness ($EI$)$_{eff}$ of such a wall element can be calculated in a semi-analytical way through Equation (1c) using the Gamma method. Following the expressions presented in [20], the fictive diagonal diameter for conventional sheathing boards (OSB or FPB) is further determined in the way that the horizontal displacement of the actual wall element is the same as the horizontal displacement of the simplified model with a fictive diagonal as schematically presented in Figure 6. Finally, the fictive diagonal diameter ($d_{fic}$) is expressed in the final analytical form, as follows:

$$A_{d,fic} = \frac{k_p \cdot L_d}{E_D \cdot \cos^2 \alpha} \tag{1a}$$

$$k_p = \frac{1}{D_p} = \left( \frac{H^3}{3 \cdot EI_{eff}} + \frac{H}{GA_s} \right)^{-1} \tag{1b}$$

$$(EI)_{eff} = E_b I_b + E_t I_t = E_b \cdot \frac{n_b \cdot t \cdot b^3}{12} + E_t \cdot \left( \frac{2 \cdot a^3 \cdot c}{12} + \frac{d^3 \cdot c}{12} + 2 \cdot \gamma_i \cdot A_t \cdot z^2 \right) \tag{1c}$$

$$d_{fic} = 2 \cdot \sqrt{\frac{A_{d,fic}}{\pi}} \tag{1d}$$

with $E_b$ and $E_t$ being the moduli of elasticity of the board and the timber. $E_D$ is the modulus of elasticity of the diagonal with the fictive cross-section of the diagonal ($A_{d,fic}$) and $L_d$ is the length of the diagonal. The horizontal stiffness of the panel is $k_p$, $D_p$ is the panel capacity, $H$ is the height of the panel, and $GA$ is the shear stiffness of the panel. The geometrical characteristics in Equation (1c) are schematically presented in Figure 7. However, it is important to point out that the mathematical model developed by [20] can be used only for sheets that are mechanically fastened to the timber frame by staples or nails. The effective stiffness $(EI)_{eff}$ is calculated using the Gamma method following the Eurocode 5 [36] expressions. The stiffness coefficient of fasteners $\gamma_y$ is defined in accordance with Eurocode 5 [36].

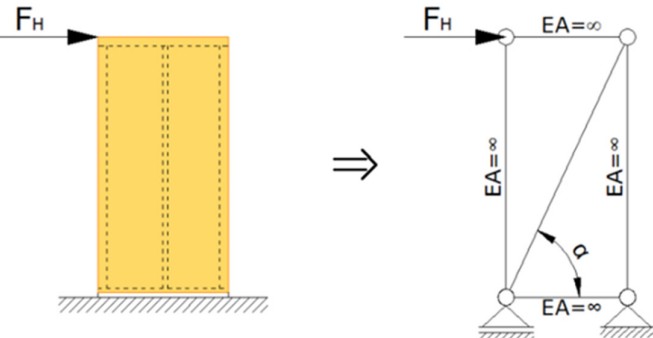

**Figure 6.** Schematic presentation of the fictive diagonal model.

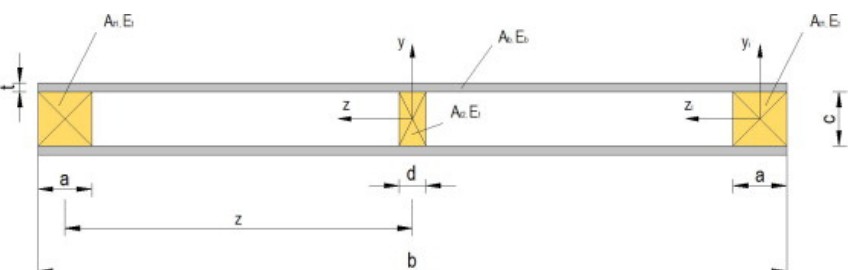

**Figure 7.** Cross-section of a typical timber-framed wall with a two-sided sheathing board [20].

*3.2. Mathematical Modelling of DSF Timber Wall Elements*

In case of timber–glass wall elements, where the glass pane is continuously bonded to the timber frame, it is not possible to determine the gamma coefficient and consequently the effective stiffness $(EI)_{eff}$ in Equations (1b) and (1c) directly with the known expressions from the Eurocodes, as it is numerically performed for the conventional sheathing boards (OSB or FPB) in Section 3.1. The diameter of the fictive diagonal ($d_{fic}$) can be determined using Equation (2):

$$d_{fic} = \sqrt{\frac{4 \cdot F_{cr} \cdot L_d}{w_{cr} \cdot (\cos \alpha)^2 \cdot \pi \cdot E_D}} \tag{2}$$

in two alternative approaches:

- By using the experimental results from [37] with the measured values for force forming the first crack in the glass pane ($F_{cr}$) and the corresponding horizontal displacement ($w_{cr}$) at the top of the wall element. However, this procedure is very expensive and also time-consuming;
- By using the numerical finite element method (FEM) approach, where the flexibility of the bonding line is simulated with elastic spring elements. This approach is briefly described below.

Certain developed mathematical finite element (FE) models with spring elements simulate the flexibility of the bonding line between the glass pane and the timber frame [24]. The model is based on an extensive numerical parametric study performed only for single-skin façade (SSF) elements and presented in [15]. The adhesive bonding of the glazing panes to the timber frame was modelled using elastic linear link elements (springs) as schematically presented in Figure 8 and introduced by Kreuzinger and Niedermaier [38]. In the computational model, the timber frame was modelled with one-dimensional finite elements (beams and studs) and the glazing panels with two-dimensional finite elements of the "composite shell" type, which allows the simulation of multilayer shells. The timber material was considered as an isotropic elastic material (with the modulus of elasticity $E_{0,mean}$) and the elements of the timber frame were modelled as the simple plane stress elements. As glass is a very brittle material, it was therefore modelled as acting linearly elastic in tension and compression; in reality, the adhesive bonding is provided continuously over the whole perimeter, and the stiffness properties of discrete spring elements were defined based on the spacing of the springs ($l_a$) in the computational model using Equations (3) and (4) for the bonding line of the inner and outer glass panes (see Figure 5) separately:

$$K_1 = \frac{E_a \cdot A_a}{t_a} = \frac{E_a \cdot (w_a \cdot l_a)}{t_a} \tag{3}$$

$$K_2 = \frac{G_a \cdot A_a}{t_a} = \frac{G_a \cdot (w_a \cdot l_a)}{t_a} \tag{4}$$

where $K_1$ is stiffness in the direction normal to the connected plane, while $K_2$ is the shear stiffness in the two perpendicular directions in the connected plane. $E_a$ and $G_a$ are the modulus of elasticity and the shear modulus of the adhesive material, respectively, $t_a$ and $w_a$ designate the thickness and the width of the adhesive layer, respectively, and $l_a$ is the impact length for a single spring element and is equal to the spacing between the springs.

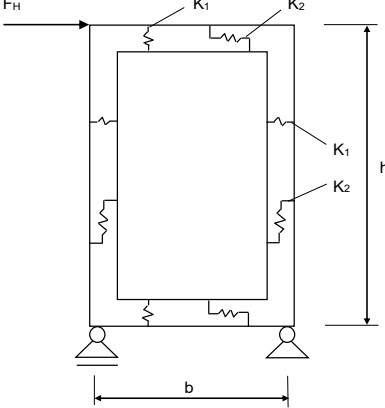

**Figure 8.** Spring model introduced by Kreuzinger and Niedermaier [38].

With the presented FE model force forming the first crack in the glass pane ($F_{cr}$), the corresponding horizontal displacement ($w_{cr}$) at the top of the wall element at this force

can be calculated. The diameter of the fictive diagonal ($d_{fic}$) can be determined now using the expression in Equation (2).

The calculated values of the diameter of the fictive diagonal element ($d_{fic}$), considering various values for the width ($w_a$) and the thickness ($t_a$) of the adhesive in the bonding line, are presented in Tables 1 and 2. The diameter of the fictive diagonal ($d_{fic}$) was first determined for different widths of the adhesive layer, considering a constant adhesive thickness of $t_a = 7$ mm (Table 1), and also for different adhesive thicknesses, considering a constant (experimental) width of the adhesive layer of $w_a = 28$ mm (Table 2). Both tables present the racking stiffness (R) of the DSF wall element calculated with the programme SAP 2000 [21] using the spring mathematical model (Figure 8) and the values for fictive diagonal diameters subsequently obtained by Equation (2).

**Table 1.** Fictive diagonal diameter and stiffness at different widths of the adhesive layer $w_a$ ($E_a$ = 1.083 MPa).

| Width of the Adhesive Layer $w_a$ (mm) | Racking Stiffness R (N/mm) | Diameter of the Fictive Diagonal $d_{fic}$ (mm) |
|---|---|---|
| 24 | 792 | 8.194 |
| 28 * | 857 | 8.521 |
| exp. | 909 | |
| 32 | 917 | 8.817 |
| 36 | 976 | 9.092 |

\* Additional experimental study performed on DSF elements [37].

**Table 2.** Fictive diagonal diameter and stiffness at different adhesive thicknesses $t_a$ ($E_a$ = 1.083 MPa).

| Adhesive Thickness $t_a$ (mm) | Racking Stiffness R (N/mm) | Diameter of the Fictive Diagonal $d_{fic}$ (mm) |
|---|---|---|
| 3 | 1563 | 11.506 |
| 5 | 1080 | 9.566 |
| 7 * | 857 | 8.521 |
| exp. | 909 | |
| 9 | 765 | 8.052 |

\* Additional experimental study performed on DSF elements [37].

A comparison of the numerical and experimental results for the analysed DSF elements is very briefly presented in Figure 9. However, a deep analysis can be found in [19].

The results show that the racking stiffness (R) increases with the increasing width of the adhesive layer ($w_a$) or the decreasing adhesive thickness ($t_a$). In the opposite case, stiffness decreases. This was expected, as a higher thickness or lower effective width of the adhesive results in the higher yield strength of the joint between the timber frame and the glass. Similar conclusions were obtained for SSF elements in the parametric numerical study in [16] and will not be further analysed in this paper.

However, in the current form, there is no further developed possible mathematical analytical or semi-analytical correlation between the spring stiffness ($K_1$ in Equation (3) and $K_2$ in Equation (4)) and the effective bending stiffness ($EI_{eff}$ in Equation (1c)) to follow the developed expressions from Equations (1a-1d) to analytically determine the needed value of the fictive diagonal parameter ($d_{fic}$), as it can be very simply analytically performed for conventional FPB or OSB sheathing boards. The DSF modelling presented here is therefore an extension of the previously developed mathematical models of this type by using spring elements (schematically presented in Figure 8) to simulate the bonding line [19]. However, at this point, it is of the utmost importance to point out that using the described FEM procedure with spring elements to determine the racking stiffness of DSF elements is very time consuming. Moreover, the calculation time is too lengthy to be

implemented into the whole structural building model for the practical engineering implementation for the seismic analysis of the whole structure.

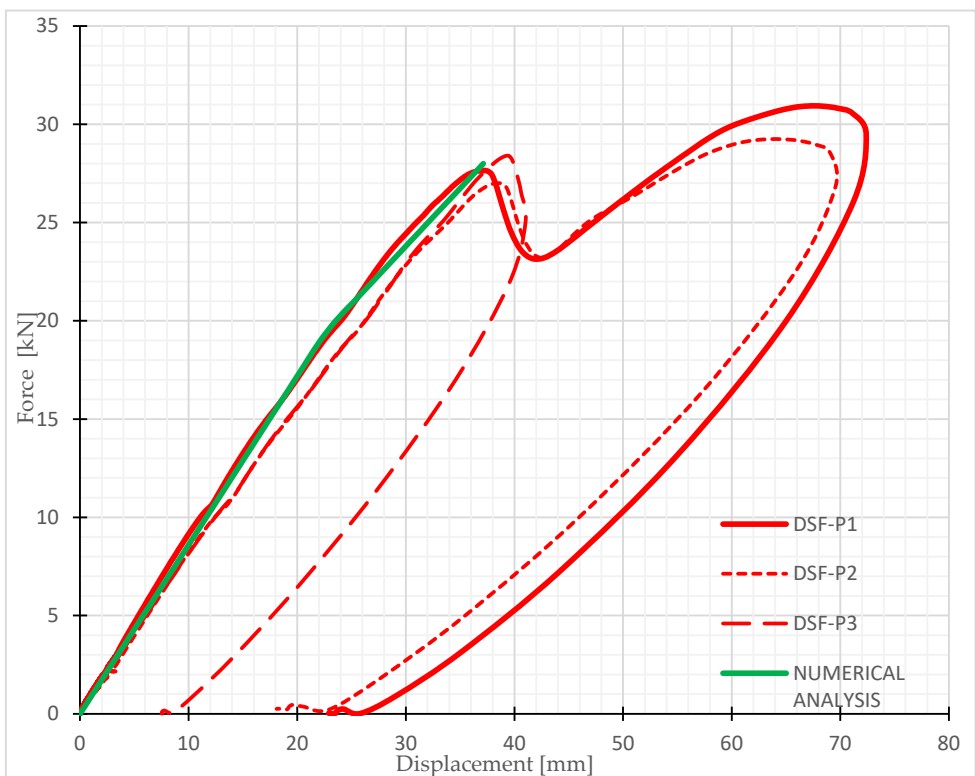

**Figure 9.** Experimental and numerical force-displacement diagrams for the DSF-P wall element.

## 4. Special Numerical Study on Selected Three-Storey Timber-Framed Building

The seismic resistance of a three-storey prefabricated building in Ljubljana (LJ), with a constant floor plan in each storey as shown in Figure 10, has been analysed. The points ABCD are the corner points on the top storey of the building, where the racking stiffnesses (R) and displacements (U) were calculated. The side view of the building is shown in Figure 11, where the marks 1-8 and A-G are the axes where the load-bearing wall elements are located. The aim of our study is to compare the seismic resistance of a three-storey building, where DSF elements will be considered as load-bearing and non-load-bearing elements, and to assess a possible contribution of the load-bearing DSF elements to the overall seismic resistance of the chosen timber building. Therefore, two alternative structural analyses were performed:

(a) The DSF elements are considered as non-resisting structural wall elements;
(b) The DSF elements are considered as horizontal-load-resisting structural wall elements.

The analyses will be focused primarily on the following two items:

(a) An increase in the overall racking stiffness of the building if the DSF elements are considered as racking resisting;
(b) A decrease in the distortional effect in the first storey if the DSF elements are considered as racking resisting.

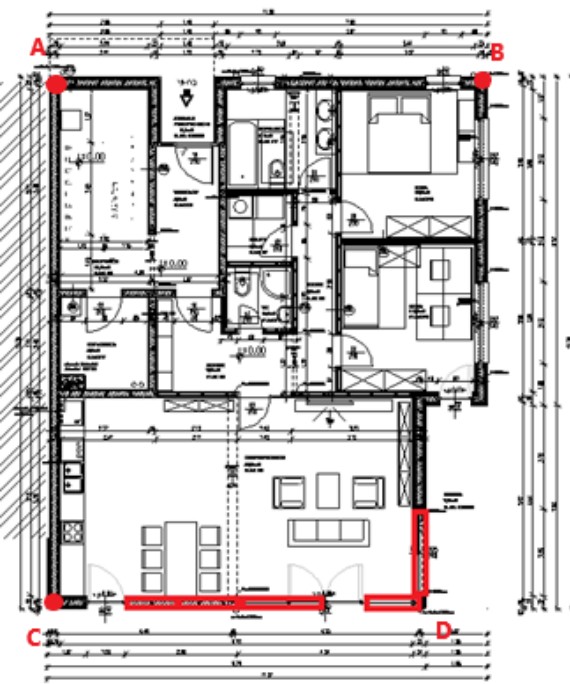

**Figure 10.** Floor plan of the building and the marked DSF elements (in red).

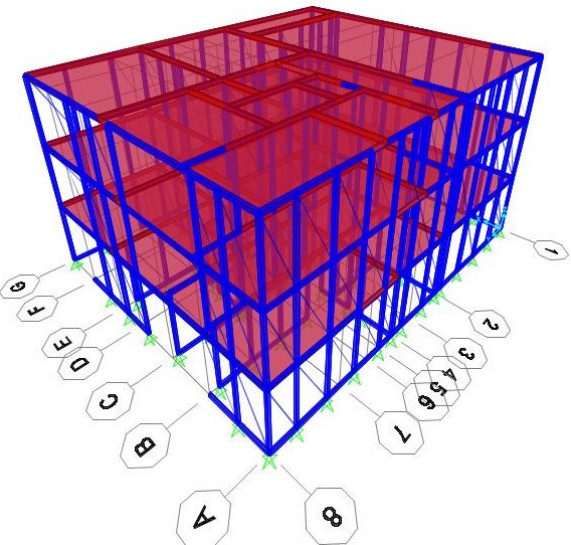

**Figure 11.** 3D model of the analysed three-storey building.

The DSF elements are primarily positioned on the south façade (direction X) to increase solar gains through the transparent areas. The seismic analysis of the building was performed using a 3D mathematical model and a modal analysis of the structure by using the commercial finite element model computer programme SAP 2000 Nonlinear v 23.0.0 [21]. Therefore, in alternative (a), only solid wall elements without any openings and DSF wall elements were considered as load-bearing elements, and in the alternative (b), DSF elements with $t_a = 7$ mm and $w_a = 28$ mm, which are marked in red on the floor plan (Figure 10), were taken into account in the calculation, using fictive diagonal elements to simulate the racking stiffness of such transparent elements.

Since the length of the wall elements is not ideal (multiples of 1.25 m), we made the assumption for all wall elements that each wall is a multiple of 1.25 m. The principle is that for each wall, the number of full wall elements (1.25 m long) is determined, and the remaining length is considered a full element if the length of the remaining part of the wall is more than half the length of the full wall, i.e., 0.625 m. All wall elements (external and internal) consist of a timber frame and two 2.50 m high sheathing boards. Fibre–plaster boards (FPB) are generally used as sheathing boards. Where the wall elements must be reinforced, OSB sheathing boards were used instead of fibre–plaster boards.

The structure was modelled using the previously described model with a fictive steel diagonal with a circular cross-section, where the stiffness of the wall element is simulated by the fictive diameter of the diagonal ($d_{fic}$). Table 3 shows the fictive diagonal diameters and the racking stiffness of each resisting wall element, the conventional sheathing boards (with FPB or OSB sheathing boards for the external and internal wall timber-framed wall elements) and the load-bearing DSF elements. Both values for the DSF element were taken from Tables 1 and 2 by using Equations (2)–(4), while the values for the OSB or FPB timber-framed wall elements were calculated according to Equations (1a-1d).

**Table 3.** Diameter of the fictive diagonals and load-bearing capacities of the wall elements.

| Timber-Framed Wall Elements | Racking Stiffness (R) of the Resisting Wall Elements (N/mm) | Diameter of the Fictive Diagonal $d_{fic}$ (mm) |
|---|---|---|
| DSF | 857 | 8.52 |
| OSB—external wall | 2800 | 16.90 |
| OSB—internal wall | 2482 | 15.93 |
| FPB—external wall | 4192 | 20.70 |
| FPB—internal wall | 3962 | 20.10 |

The supports are also simulated as fully rigid. At the same time, the axial stiffness of the frame in the calculation model should be high enough so that the influence of the frame ductility can be eliminated and only the ductility of the diagonals can be considered [20]. The material used for the diagonals was steel with a modulus of elasticity of E = 210 GPa. The surface load subjected to the floor slabs was 2 kN/m². A computational model has been carried out for a three-storey timber building with identical floor plan dimensions. Figure 11 shows the 3D model for the three-storey building, while Figure 12 shows the model of the wall in Axis 1 using a load-bearing and non-load-bearing DSF element.

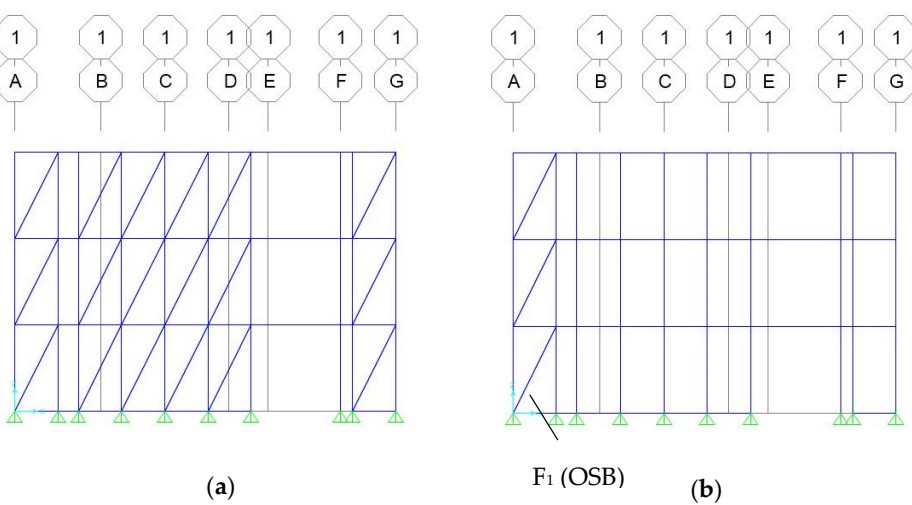

(**a**)　　　　　　　　　　F₁ (OSB)　(**b**)

**Figure 12.** View of Axis 1 of the three-storey wall model using load-bearing (**a**) and non-load-bearing DSF elements (**b**).

## 5. Discussion of Results

The numerical results for the three-storey prefabricated building have been compared, considering the DSF elements as load-bearing (alternative (b)) and non-load-bearing (alternative (a)). In the case of the load-bearing DSF elements, the thickness of the fictive diagonal was 8.52 mm. The comparison was performed for the location in Ljubljana, where the design ground acceleration ag is 0.25 g. First, the oscillation times of the structure were calculated. Table 4 shows the oscillation times (the first three oscillation modes) of the structure for the two analysed alternatives with resisting and non-resisting DSF elements.

**Table 4.** Oscillation times of the three-storey building considering load-bearing and non-load-bearing DSF wall elements.

| DSF Element | Non-Load-Bearing DSF Elements (a) | Load-Bearing DSF Elements (b) |
|---|---|---|
| Oscillation Mode | Oscillation Times T [s] | Oscillation Times T [s] |
| 1. | 0.479 | 0.438 |
| 2. | 0.367 | 0.363 |
| 3. | 0.308 | 0.303 |

As expected, the oscillation times are higher when considering non-load-bearing DSF elements, because the racking stiffness is in this case smaller and the mass is supposed to be unchanged. The calculated horizontal (racking) stiffnesses (R) and displacements of the structure (at the resultant force $F_H$ = 376 kN for non-load-bearing DSF elements and $F_H$ = 386 kN for load-bearing DSF elements) in both global orthogonal directions of the earthquake action (directions X and Y) for the individual selected control points (A–D), considering both load-bearing and non-load-bearing DSF elements, are shown in Table 5.

**Table 5.** Racking stiffnesses (R) and displacements (U) of the corner points on the top storey of the three-storey building.

| DSF Element | | Load-Bearing DSF Elements | | Non-Load-Bearing DSF Elements | |
|---|---|---|---|---|---|
| Earthquake | | Direction X | Direction Y | Direction X | Direction Y |
| Location | | LJ | | LJ | |
| Point | Displacement (mm) | | | | |
| A | $U_x$ | 10.25 | 7.96 | 9.35 | 8.51 |
| | $U_y$ | 1.09 | 6.37 | 2.31 | 6.89 |
| | $U_R$ | 10.31 | 10.20 | 9.63 | 10.95 |
| B | $U_x$ | 10.25 | 7.95 | 9.35 | 8.51 |
| | $U_y$ | 7.87 | 10.90 | 11.13 | 10.44 |
| | $U_R$ | 12.92 | 13.49 | 14.54 | 13.47 |
| C | $U_x$ | 17.35 | 6.59 | 21.64 | 7.56 |
| | $U_y$ | 1.09 | 6.38 | 2.32 | 6.90 |
| | $U_R$ | 17.38 | 9.17 | 21.76 | 10.24 |
| D | $U_x$ | 17.35 | 6.59 | 21.64 | 7.56 |
| | $U_y$ | 7.88 | 10.91 | 11.14 | 10.45 |
| | $U_R$ | 19.06 | 12.75 | 24.34 | 12.90 |
| R (N/mm) | | 15,198 | 18,859 | 11,314 | 18,953 |

The allowed value of the horizontal displacements for a multi-storey building, according to [22] is H/500, which, in our case, amounts to 15 mm. For Ljubljana, the values of horizontal displacements exceed the prescribed Eurocode limits by about 21% when

DSF are considered as resisting and by about 38% when DSF are considered as non-resisting. These values are marked in red in the table. The overall racking stiffness (R) of the whole building also essentially increases if load-bearing DSF elements are used, i.e., by 34% in the X direction, while there is almost no influence for the Y direction because almost all load-bearing DSF elements are placed in the X direction (south face) of the building façade only. Therefore, the use of load-bearing DSF elements is very important in this case.

Table 6 shows acting horizontal forces ($F_x$ and $F_y$, respectively) due to seismic action in two global perpendicular directions (directions X and Y) and the resulting force (FR) in the external walls of the building façade (axes 1, 8, A, G), which are shown in Figure 11. The horizontal force acting on the corner timber-framed wall element 1 in Axis 1 with the conventional OSB sheathing boards ($F_1$ marked in Figure 12b) is particularly controlled to assess the influence of the DSF load-bearing elements in decreasing the distortion of the first floor.

**Table 6.** Horizontal forces in the exterior walls of the building.

| DSF Element | | Load-Bearing DSF Elements | | Non-Load-Bearing DSF Elements | |
|---|---|---|---|---|---|
| Earthquake | | Direction X | Direction Y | Direction X | Direction Y |
| Location | | LJ | | LJ | |
| Axis | Force (kN) | | | | |
| 1 | $F_x$ | 52.82 | 21.67 | 30.77 | 11.55 |
| | $F_y$ | 126.01 | 64.38 | 130.11 | 58.53 |
| | $F_R$ | 136.63 | 67.93 | 133.70 | 59.66 |
| 8 | $F_x$ | 43.78 | 43.62 | 39.44 | 45.94 |
| | $F_y$ | 80.13 | 119.42 | 117.79 | 120.68 |
| | $F_R$ | 91.31 | 127.14 | 124.22 | 129.13 |
| A | $F_x$ | 13.68 | 76.43 | 31.71 | 83.51 |
| | $F_y$ | 161.29 | 138.69 | 183.43 | 147.10 |
| | $F_R$ | 161.87 | 158.36 | 186.15 | 169.15 |
| G | $F_x$ | 28.80 | 47.96 | 42.92 | 47.38 |
| | $F_y$ | 130.93 | 128.20 | 131.08 | 125.20 |
| | $F_R$ | 134.06 | 136.88 | 137.93 | 133.87 |
| Axis 1: $F_1$ (kN) | | 20.20 | 24.76 | 24.56 | 24.68 |

It is evident from the presented results that the force $F_1$ decreased notably (by almost 18%) when the load-bearing DSF elements were used for seismic excitation in the X direction. On the other hand, there is practically no influence for seismic excitation in the Y direction, as similar results were observed for the overall racking stiffness of the building in Table 5. Consequently, it can be concluded that the distortion effect on the building can be essentially decreased by using load-bearing DSF elements.

## 6. Conclusions

Based on the computational analyses performed, the use of the developed DSF elements as load-bearing structural elements to increase the racking load-bearing capacity of the whole structure proved to be very reasonable, since the racking stiffness of the entire building could be increased by up to 35% with the given fixed installation of the DSF elements. It should be noted that the influence of the stiffness of the DSF elements on the overall racking stiffness of the whole building depends on the floor layout of the load-bearing DSF elements on each individual floor. Additionally, as a secondary goal of the study, it is important to point out that by using the DSF elements as racking resistant, the distortion on the first floor essentially decreased because the horizontal action in the

checked corner timber-framed wall elements with an OSB sheathing board in Axis 1 was lower by almost 18%.

The development of such racking-resistant timber DSF elements can open many new perspectives in designing contemporary multi-storey timber buildings located in seismic areas with strong winds and with a strong asymmetrical position of the transparent glass areas. On the other hand, this decreases the energy demand for heating and provides better daylight, contributing to better living comfort in the building. However, this opens many structural problems, especially for mid- and high-rise prefabricated timber buildings, which can be solved by the developed lateral resistant DSF elements. The results of the study can also have important socioeconomic effects, as they can significantly contribute to the additional expansion of multi-storey timber buildings throughout Europe, resulting in the better use of forested areas, and thus significantly contributing to the reduction of the environmental impacts of buildings.

However, there are still some problems with the mathematical modelling of resisting DSF elements using a simplified fictive diagonal model, which is only acceptable for a practical and rapid engineering static and dynamical analysis of multi-storey prefabricated timber buildings. In the current stage, the fictive diagonal model can be widely used only for conventional timber-framed wall elements with OSB or FPB boards. However, for load-bearing DSF elements, the effective bending stiffness of the composite wall elements still cannot be calculated in a semi-analytical way through Equation (1c) using the Gamma method. Therefore, the results of experimental tests or the FEM model with spring elements have to be used first to determine the diameter of the fictive diagonal using Equation (2). However, the experimental approach is very expensive and the FEM approach with springs presented in this paper is still very time consuming. Therefore, it is of the utmost importance for further research to develop a semi-analytical approach to determine the fictive diagonal diameter.

**Author Contributions:** Conceptualization, M.P. and E.K.Š.; methodology, M.P. and E.K.Š.; software, E.K.Š.; validation M.P. and E.K.Š.; formal analysis, M.P. and E.K.Š.; investigation M.P. and E.K.Š.; resources, M.P. and E.K.Š.; data curation, M.P. and E.K.Š.; writing—original draft preparation, M.P. and E.K.Š.; writing—review and editing, M.P. and E.K.Š.; visualization, M.P. and E.K.Š.; supervision, M.P.; project administration, E.K.Š. All authors have read and agreed to the published version of the manuscript.

**Funding:** Funding for this research was provided by the Slovenian Research Agency and the Ministry of Higher Education, Science and Technology of the Republic of Slovenia, National research program P2-0129 and the investment is co-financed by Republic of Slovenia and the European Union under the European Regional Development Fund (research core funding Development of multifunctional climatically active building wrapper—HOME+).

**Institutional Review Board Statement:** Not applicable.

**Informed Consent Statement:** Not applicable.

**Data Availability Statement:** Data available on request.

**Conflicts of Interest:** The authors declare no conflicts of interest.

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
