# Peer review of "Numerical Analysis of the Racking Behaviour of Multi-Storey Timber-Framed Buildings Considering Load-Bearing Function of Double-Skin Façade Elements"

_sustainability, doi:10.3390/su15086379_

Round 1
Reviewer 1 Report
This manuscript employed numerical analysis in a comparative study to understand the racking behaviour of multi-storey timber-framed buildings with load-bearing double-skin façade elements. The reviewer recommends it to be considered for publication on this journal after some minor revisions.
1. The title of this article should be considered again. It should completely summarize the content of this article.
2. In the introduction, references should be written in chronological order.
3. L#10- “façade” in “double-skin façade elements” does not use standard language.
4. L#12- you speak about “OSB”, but you never describe it. People may not know what those letters mean.
5. References should be added here “It is generally known that wind loads exponentially increase only from a certain height of a building upwards, while earthquake loads increase almost linearly with the height of a building as schematically shown in Figure 3 for cases when higher loads are caused by earthquake loads.”
6. All letters in the formula need to be defined.
7. The quality of the picture should be increased, for example, the labels in Figure 9 and 10 are coincident, etc.
8. The quality of tables should also be increased, and the format of Table 3 and Table 1 is obviously inconsistent.
10. It is suggested to add experimental content or compare with existing studies
11. The description is jumbled, suggest refining their own expression.
12. The article is not full enough.
Author Response
Dear Reviewer,
In the document you will find the answers and changes based on your recommendations. The article was also reviewed by an English proofreader.
Kind Regards,
Authors

Reviewer 2 Report
Author has presented the modelling of multi-storey timber framed buildings using FEM model. The work needs significant improvements, following are few recommendations
1) why there is need to to consider the double-skin façade elements? justify
2) The abstract needs to improved by adding few qualititative numbers from the study about innovative steps and results obtained
3) introduction is well written, however include some parallel studies with respect to the DSF
4) is figure 2, 3 and 4,5 are from some literature references?
5) is it possible to present results in term of figures or contours ? if yes add it
6) is there any validation study is conducted?
7) what are the assumptions made during the FEM study
8) results discussion need in details discussion on results and FEM modelling
9) conclusion can be improved by mentioning key findings and adding future work
Author Response

(The authors gave the same response as above.)

Reviewer 3 Report
This paper proposes a method for modeling multi-storey timber-frame buildings, double-skin façade (DSF). Although a lot of work has been done in this article, there are still many unclear places in terms of text expression and research ideas. Therefore, major revisions are needed to realize the true value of research results. The following issues also need to be improved.
1. The introduction of this article is complicated and the text is too cumbersome, and it is recommended to delete the common-sense explanatory text without changing the main content of the article to make the article more concise.
2. Figure 2 “Horizontal distortion of a building due to antisymmetric position of resisting wall elements”does not show whether a horizontal displacement has occurred.
3. The article itself uses numerical simulation to describe the shelf performance of double-storey cubic multi-storey timber structure buildings, but some graphics (as shown in Figure 8) are not clear, and it is recommended to adjust the clarity of the picture to enable readers to better understand the article.
4. The data in the article should be accurate and detailed, such as section 2.1 to explain the current wooden building form to achieve the maximum, what does it mean, if it is a specific height, please specify the specific value.
5. As we all know, wooden structures generally have limited building height and performance, so the research field is not extensive, since the article is a numerical simulation of wooden structures, why does it not explain the impact of the article on the field? Such as the impact on industrial production and so on.
6. The article has cited fewer literature in the past five years, and the citation of the article should keep pace with the times, citing more internationally renowned journals and journals with a publication period of the past five years.
7. In the article, when formula symbols first appear in manuscripts, the author should clearly explain the representation of the symbols. For example, the symbols kp, Ib, etc. in equation (1) are not stated. Since there are many variables in the text, it is recommended that the meaning of each symbol should be explained.
8. If the formula in the article is not self-derived and is not a common formula, add a quote to explain it.
Author Response

(The authors gave the same response as above.)

Round 2
Reviewer 2 Report
no further commnets
Reviewer 3 Report
The author has made reasonable modifications to the reviewer's comments.